# Roles of Neuropeptides in Sleep–Wake Regulation

**DOI:** 10.3390/ijms23094599

**Published:** 2022-04-21

**Authors:** Yi-Chen Shen, Xiao Sun, Lei Li, Hu-Yunlong Zhang, Zhi-Li Huang, Yi-Qun Wang

**Affiliations:** 1State Key Laboratory of Medical Neurobiology, MOE Frontiers Center for Brain Science, Department of Pharmacology, School of Basic Medical Sciences, Institutes of Brain Science, Fudan University, Shanghai 200032, China; 18301020058@fudan.edu.cn (Y.-C.S.); 18211010070@fudan.edu.cn (X.S.); 19111010089@fudan.edu.cn (L.L.); 18301050159@fudan.edu.cn (H.-Y.Z.); 2School of Public Health, Fudan University, Shanghai 200032, China

**Keywords:** neuropeptides, NREM, REM, sleep, wake

## Abstract

Sleep and wakefulness are basic behavioral states that require coordination between several brain regions, and they involve multiple neurochemical systems, including neuropeptides. Neuropeptides are a group of peptides produced by neurons and neuroendocrine cells of the central nervous system. Like traditional neurotransmitters, neuropeptides can bind to specific surface receptors and subsequently regulate neuronal activities. For example, orexin is a crucial component for the maintenance of wakefulness and the suppression of rapid eye movement (REM) sleep. In addition to orexin, melanin-concentrating hormone, and galanin may promote REM sleep. These results suggest that neuropeptides play an important role in sleep–wake regulation. These neuropeptides can be divided into three categories according to their effects on sleep–wake behaviors in rodents and humans. (i) Galanin, melanin-concentrating hormone, and vasoactive intestinal polypeptide are sleep-promoting peptides. It is also noticeable that vasoactive intestinal polypeptide particularly increases REM sleep. (ii) Orexin and neuropeptide S have been shown to induce wakefulness. (iii) Neuropeptide Y and substance P may have a bidirectional function as they can produce both arousal and sleep-inducing effects. This review will introduce the distribution of various neuropeptides in the brain and summarize the roles of different neuropeptides in sleep–wake regulation. We aim to lay the foundation for future studies to uncover the mechanisms that underlie the initiation, maintenance, and end of sleep–wake states.

## 1. Introduction

Sleep is one of the most important physiological functions in mammals. It is often described as the normal loss of consciousness. The main function of sleep is to eliminate fatigue. Sleep is also involved in the process of learning and memory consolidation. As a result, sleep disorders have a negative effect on physical and mental health. In terms of characteristics of the electroencephalogram (EEG), sleep in mammals can be divided into two distinct stages: rapid eye movement (REM) sleep and non-REM (NREM) sleep. Judging by the EEG, NREM sleep in humans can be further divided into four stages: stage 1, stage 2, stage 3 and stage 4. REM sleep, which is also called paradoxical sleep, is defined by REM, the entire absence of muscle tone, and the ability to dream vividly [1].

The sleep–wake cycle is primarily modulated by circadian rhythms and homeostatic regulation. Several specific brain regions are involved in the regulation of the sleep–wake cycle, including the forebrain, hypothalamus, and brain stem. Previous studies have revealed that the nuclei involved in the regulation of arousal response include the thalamus, basal forebrain (BF) [2], lateral hypothalamus (LH) [3], tuberomammillary nucleus (TMN) [4], ventral tegmental area [5,6], the pedunculopontine (PPT)/laterodorsal tegmental nucleus (LDT) [7], dorsal raphe nucleus (DRN) [8], parabrachial nucleus [9], and locus coeruleus (LC) [10] (Figure 1). The nuclei associated with NREM sleep regulation include those of the olfactory tubercles [11], ventrolateral periaqueductal gray (vlPAG) [12], preoptic area (POA) [13], rostromedial tegmental nucleus [14], thalamic reticular nucleus [15], and other regions of the brain (Figure 2). The olfactory bulb [16], POA [17], vlPAG [12], PPT/LDT [7], DRN [18], and LC [19] have been reported to be involved in the regulation of REM sleep (Figure 3) [20,21].

In mammals, neuropeptides, which are engaged in many physiological functions, are the most diverse class of signaling molecules in the brain. In the mammalian genome, there are almost 70 genes encoding the bioactive neuropeptides and neuropeptide precursors [22]. Neuropeptides are small protein molecules composed of 3–100 amino acid residues [23]. Neuropeptides are synthesized from neuropeptide precursors that require proteolytic processing primarily within secretory vesicles. Mature neuropeptides are stored in these vesicles and secreted to modulate the activity of target cells [24]. Neuropeptides can be released from all parts of a neuron, including the axon, the soma, and especially the dendrite, which can be located in the central nervous system (CNS) or peripheral nervous system (PNS). After the secretion of neuropeptides, most of them bind to G protein-coupled receptors (GPCRs) and then elevate the intracellular Ca^2+^ concentrations to alter membrane excitability, transcription, and synaptogenesis, thus regulating a wide range of behaviors, including sleep–wake behaviors [25]. In addition, proteinases play a critical role in the regulation of the biological activity of neuropeptides in the CNS through proteolytic conversion and degradation. These enzymes, in turn, are regulated by inhibitors, which are involved in the regulation of many metabolic pathways [26].

Sleep is one of the most basic physiological functions in humans, yet the neurobiological mechanisms of sleep–wake regulation remain poorly understood. Several neuropeptides have been reported to be involved in the regulation of the sleep–wake cycle. In this review, we will summarize the distributions and functions of different neuropeptides in the regulation of sleep–wake behaviors.

## 2. The Neuropeptides Involved in Sleep–Wake Regulation

### 2.1. Galanin Promotes Sleep

Galanin was discovered by Mutt’s team at the Karolinska Institute in Stockholm in the 1980s. It is composed of 29 amino acids (30 in humans) and is considered as a “classical neuropeptide” that regulates neurotransmission in the CNS and PNS. Galanin has been reported to play a significant role in the regulation of numerous physiological and pathophysiological processes through the interaction with three GPCRs [27,28,29], including feeding [30], energy homeostasis [31], osmotic regulation [32], water intake [33], and pain [34]. In addition, the mRNA expression of galanin is found in γ-aminobutyric acid (GABA)-positive neurons in the ventrolateral preoptic nucleus (VLPO) [35], which is a critical nucleus of the sleep–wake regulation. It indicates that galanin is involved in the regulation of VLPO activity. Moreover, several studies have demonstrated that the VLPO sends inhibitory projections to many sleep-related nuclei, such as the TMN and other arousal systems in the brain stem, including the DRN and the LC [36]. As a result, galanin affects sleep–wake behaviors.

Previous studies have revealed that galanin promotes sleep. In a clinical trial, intravenous administration of galanin tended to increase the duration of REM sleep in young men [37]. In another study, the overexpression of galanin reduced spontaneous locomotor activity and responses to sensory stimulation in zebrafish [38]. In addition, Kroeger et al. found that the neurons in the VLPO, which co-express GABA and galanin, express c-Fos after periods of increased sleep. In the same study, optogenetic inhibition of the galanin-expressing neurons in the VLPO impeded sleep in rats, while chemogenetic activation of these neurons resulted in an increase in sleep duration and a decrease in sleep latency [39]. Another study has illustrated that galanin promotes sleep by inhibiting the noradrenergic system which induces wakefulness. Specifically, galanin reduced the activity of noradrenergic neurons in the LC [40]. All these findings indicate that galanin has a positive impact on sleep.

Studies on the mechanisms of galanin-related sleep–wake regulation are scarce. Most of the relevant studies have targeted galanin-expressing neurons in the VLPO, but galanin is widely distributed in the CNS. Therefore, it is our opinion that research in this area is still lacking.

### 2.2. Orexin Consolidates Wakefulness and Inhibits REM Sleep

Orexin exists in two molecular forms, orexin-A and orexin-B, which both perform physiological functions through the interaction with the GPCRs [41]. Rats have around 3000–4000 orexin-producing neurons in the brain [42], and they are located mainly in the peripherical area of the LH [43,44]. These neurons project widely to the CNS, regulating feeding and other behaviors. Many nuclei that regulate sleep–wake behaviors receive projections from the orexin neurons as well, including the LC [45], DRN [46], TMN [47], PPT/LDT [48], and BF [49]. Therefore, orexin is involved in sleep–wake regulation.

Several studies have proved that orexin induces wakefulness and inhibits REM sleep [41,50]. Mieda et al. revealed that the intracerebroventricular administration of orexin-A in rodents suppressed both NREM and REM sleep and increased the duration of wakefulness significantly [51]. In addition, application of an orexin receptor-2 antagonist promoted NREM and REM sleep [52]. Another experiment also found that optogenetic activation of orexin neurons accelerated the transition from sleep to wakefulness [53]. Mignot et al. have reported that narcolepsy, a typical type of sleep disorder, is caused by orexin deficiency [54].

The direct administration of orexin-A into the nuclei that receive projections from orexin neurons (such as LC [55], TMN [56], DRN [46], BF [57], and LDT [58]) has also been reported to increase the duration of wakefulness. Moreover, for in vitro slice electrophysiology studies, some researchers found that orexin-A and orexin-B increased the firing rates of the monoaminergic neurons in the LC [59,60], DRN [60,61], and TMN [47,62], and the cholinergic neurons in the BF and LDT [63,64]. Optogenetic activation of orexin neurons increased the activity of the neurons in the LC [45]. These observations suggest that orexin neurons stabilize wakefulness by regulating the monoaminergic and cholinergic neurons in the downstream nuclei.

In addition, orexin has been reported to suppress the NREM- and REM-promoting regions. One experiment indicated that the decrease in orexin disinhibits the REM-sleep-related regions, such as the sub-LDT and potentially the neurons in the LDT/PPT, which consequently promotes REM sleep [65]. These studies suggest that orexin neurons play a significant part in modulating REM sleep.

Orexin neurons have also been reported to receive projections from some nuclei involved in sleep–wake regulation. The GABAergic neurons in the POA, including VLPO, densely innervate the orexin neurons [66]. Orexin neurons are strongly inhibited by both the GABA_A_ receptor agonist muscimol and the GABA_B_ receptor agonist baclofen [67], thus influencing REM sleep. Optogenetic stimulation of the POA fibers obviously reduces the activity of orexin neurons [68]. In addition, Mieda et al. have proved that the 5-HT neurons in the DRN send inhibitory projections to orexin neurons as well [67,69]. A study on mice lacking 5-HT receptors showed an increase in REM sleep [41]. Another study established that ablation of all 5-HT neurons in the CNS decreased the duration of REM sleep and attenuated the arousal response [70]. In other words, 5-HT neurons are beneficial to maintaining the architecture of wakefulness and REM sleep. In this way, the 5-HT input to orexin neurons plays an important role in regulating REM sleep and wakefulness [41] and the noradrenergic neurons also have an inhibitory effect on orexin neurons [67,71]. Moreover, orexin neurons are also innervated by the cholinergic neurons in the BF, which have positive influence on wakefulness [72]. Carbachol, an agonist of muscarinic receptors, activates a subset of orexin neurons [66]. Overall, orexin neurons are inhibited by sleep-promoting neurons and activated by wake-promoting neurons. The negative feedback is crucial for the maintenance of wakefulness and the onset of REM sleep [73].

Local feedback circuits also play an important role in the regulation of orexin neurons. Orexin neurons activate themselves directly and indirectly via the local glutamatergic neurons, forming a positive feedback circuit that may stabilize the activity of the orexin neuron network [74]. However, orexin also activates the local GABAergic input to orexin neurons. Selective elimination of GABA_B_ receptors on orexin neurons has been reported to cause the fragmentation of sleep–wake states, which disrupts inhibitory signals from local GABAergic neurons to orexin neurons [75].

The large number of studies on orexin above suggest that orexin plays an important role in promoting wakefulness and inhibiting REM sleep. We have not only determined the effects of direct administration of orexin or activation of orexin neurons on sleep–wake behaviors, but also explored the role of upstream and downstream nuclei of orexin neurons in sleep–wake regulation. Nevertheless, the neurobiological mechanisms by which orexin affects sleep–wake regulation need to be investigated further.

### 2.3. Melanin-Concentrating Hormone Has Positive Influence on Sleep

Melanin-concentrating hormone (MCH) is a cyclic neuropeptide consisting of 19 amino acids [76], which performs physiological functions through the interaction with two GPCRs known as MCH receptor-1 and MCH receptor-2. It serves as an important neuromodulator of homeostasis and performs a large range of integrative functions, which are mainly associated with homeostatic regulation and motivated behaviors [77,78]. In mammals, MCH neurons are mainly located in the LH and the zona incerta [79,80]. They project to many nuclei that promote REM sleep and arousal, including the LC, DRN, LDT/PPT, and the sub-LDT [81]. Although the location and projection of MCH neurons are remarkably similar to those of orexin neurons [81], they have opposite effects on the modulation of sleep–wake states. MCH neurons have a positive influence on sleep, especially REM sleep [76].

There is much evidence supporting the sleep-promoting effect of MCH. Studies on MCH knockout mice have shown a decrease in slow-wave sleep and an increase in wakefulness. The administration of MCH has been found to increase the duration of NREM and REM sleep in rats [82]. Moreover, the intracerebroventricular administration of an MCH receptor-1 antagonist has been reported to reduce NREM and REM sleep in rats [83]. Additionally, a recent study demonstrated that optogenetic activation of MCH neurons for 24 h increased both NREM and REM sleep [84]. It has also been reported that selective optogenetic activation of MCH neurons during NREM sleep facilitates the transition from NREM sleep to REM sleep, while optogenetic activation of MCH neurons during REM sleep stabilizes and extends the duration of REM sleep episodes [85]. These studies indicate that MCH plays an important part in inducing sleep, especially REM sleep.

However, the downstream pathways through which MCH neurons modulate sleep–wake behaviors remain unclear. Lagos et al. found that MCH administration into the DRN could produce a dose-dependent increase in the duration of REM sleep as well as a modest increase in the duration of slow-wave sleep in rats [86], whereas MCH administration into the VLPO neurons increased NREM sleep [87]. MCH neurons have also been found to inhibit the vlPAG and the lateral pontine tegmentum, which suppress REM sleep. Chemogenetic activation of MCH terminals in the vlPAG and the lateral pontine tegmentum tends to increase the duration of REM sleep [88]. Overall, MCH neurons promote sleep by inhibiting the wakefulness-promoting nuclei [85].

Regarding the sleep-promoting effect of MCH, different studies have shown different results regarding the sleep–wake stages it facilitates, which needs to be further verified by more experiments. The downstream nuclei projected from MCH neurons have also been insufficiently studied, and the mechanisms by which MCH regulates sleep–wake states remain unknown.

### 2.4. Neuropeptide S Is Associated with Arousal Induction

Neuropeptide S (NPS) is a peptide composed of 20 amino acids and an endogenous ligand for the NPS receptor. The NPS receptor is a typical GPCR, containing seven membrane-spanning domains. The N-terminal residue of NPS in all species is always serine, hence the name NPS [89]. NPS is expressed in the brainstem, amygdala, hippocampus, and in other regions of the limbic system. Rainer et al. have demonstrated that mRNA expression of the NPS receptor is widespread throughout the CNS, and it is especially abundant in the cortex, thalamus, hypothalamus, and amygdala [90]. In addition, low mRNA levels of the NPS receptor have been observed in the brainstem. In contrast, mRNA expression of the NPS precursor is mainly observed in brainstem nuclei such as the LC and the lateral parabrachial nucleus, while a small number of scattered NPS-positive neurons are found in other brain areas, such as the amygdala and hypothalamus [89,91]. NPS induces the mobilization of intracellular Ca^2+^ [92], increases the intracellular cAMP levels, and stimulates the phosphorylation of mitogen-activated protein kinase [93]. NPS modulates a variety of physiological functions such as food intake [94,95,96], the regulation of the endocrine system, spatial memory, alcohol seeking, nociception, and anxiety.

NPS has been reported to exert a dual arousal and anxiolytic effect in rodents. NPS agonists are clinically applied to treat hypersomnia and anxiety disorders, while NPS antagonists might serve as novel therapeutic tools to treat insomnia. Intracerebroventricular administration of NPS tends to cause a significant increase in locomotor activity [97] and wakefulness duration [98]. Furthermore, a study demonstrated that infusion of NPS into the bilateral anterior hypothalamus, which includes the VLPO, increases the duration of wakefulness episodes significantly, and specifically decreases NREM sleep [99]. It indicates that NPS induces wakefulness by suppressing the sleep-promoting neurons in the VLPO.

In summary, the in-depth study of NPS and its receptors is beneficial for the treatment of insomnia and anxiety disorders. However, there is a lack of research on the neurobiological mechanisms by which NPS regulates sleep–wake behaviors.

### 2.5. Neuropeptide Y Has a Dual Impact on Sleep–Wake Behaviors

Neuropeptide Y (NPY) is a highly conserved endogenous peptide consisting of 36 amino acids, which is widely distributed in the CNS and PNS of all mammals and acts as a neurohormone and neuromodulator. NPY exerts its biological functions via the interaction with five subtypes of GPCRs [100]. NPY and its receptors are involved in a variety of behaviors, such as food intake, circadian rhythms, chronic pain, the stress response, and anxiety [101,102,103,104,105,106]. NPY is expressed in many nuclei associated with sleep–wake regulation, such as the amygdala, hypothalamus, hippocampus, periaqueductal gray, LC, and the cerebral cortex [107,108,109]. Therefore, NPY is thought to affect sleep–wake behaviors [110].

Several studies have proved that NPY promotes sleep. For example, NPY could shorten the sleep latency in young men [110]. Huang et al. also observed that the morning plasma NPY levels of patients with primary insomnia were significantly lower than those in the control group, suggesting low NPY levels may contribute to the pathophysiological process of primary insomnia [111]. Animal studies have also reported that the intraventricular injection of NPY enhanced EEG synchronization and increased sleep duration [112]. In addition, loss of NPY or NPY-expressing neurons resulted in less daytime sleep [113]. A previous study also found that the application of PYY3-36, an NPY receptor-2 agonist, increased the duration of NREM sleep in rats [114], but the neurobiological mechanisms by which NPY induces sleep remain unknown. Singh et al. have demonstrated that NPY promotes sleep by inhibiting the wakefulness-promoting noradrenergic neurons [113]. Moreover, NPY is reported to inhibit the activity of orexin neurons by multiple presynaptic and postsynaptic mechanisms, indicating that NPY axons have a negative effect on the orexin-regulated arousal response [115].

On the contrary, Stanley et al. found that the direct injection of NPY into the paraventricular nucleus decreases the sleep duration [103]. When rats begin to sleep and increase their VLPO activity, NPY administration into the LH suppresses NREM and REM sleep in rats when injected at the onset of the light phase [116]. Thus, NPY also tends to induce arousal, which may be explained by the inhibition of VLPO neurons, which express NPY receptor-1 [117,118].

In conclusion, NPY has been reported to either induce or impede sleep. The reasons accounting for the paradoxical phenomenon are still unknown and need to be further studied. There is also a lack of research into the mechanisms.

### 2.6. Substance P Induces Either Sleep or Arousal

Substance P (SP) is a neuropeptide consisting of 11 amino acids [119]. It exerts its functions by binding to neurokinin receptors, particularly neurokinin type 1 receptors (NK-1Rs) and NK-2Rs [120]. NK-1Rs in the CNS are critical for the regulation of affective behaviors, neurochemical responses to stress, and pain transmission [121,122]. NK-1Rs are distributed throughout the CNS, including many brain regions that are highly involved in sleep–wake regulation, such as the hypothalamus, brainstem, and cortex [123,124]. Therefore, SP affects sleep–wake states.

SP is reported to induce sleep. For example, bilateral microinjection of SP into the VLPO increases NREM sleep in rats [125]. Microinjection of SP into the cerebral cortex enhances the slow-wave activity in mice as well [126]. Furthermore, a previous study has shown that the intracerebroventricular administration of SP conjugated with cholera toxin A subunit can enhance NREM sleep but induce sleep fragmentation [127]. However, on the contrary, the systemic administration of non-nociceptive doses of SP has been reported to increase the duration of wakefulness episodes in mice [128], and to increase the latency of REM sleep and wakefulness in healthy young men [129]. Sergeeva et al. have also found that SP induces arousal by activating histaminergic neurons in the TMN [130].

These results suggest that the activation of SP-expressing neurons in the brain tends to produce not only wakefulness-promoting but also sleep-inducing effects. This paradoxical phenomenon may result from the complexity of SP functions, which include increasing or decreasing the excitability of neurons [131] and the desensitization of NK-1Rs [132]. It is necessary to explore the mechanisms by which SP modulates sleep–wake behaviors in the future.

### 2.7. Vasoactive Intestinal Peptide Promotes REM Sleep

Vasoactive intestinal peptide (VIP), a peptide consisting of 28 amino acids, is produced in many regions of the human body, including the gut, pancreas, and suprachiasmatic nucleus (SCN) [133]. The SCN is the center of the circadian rhythm, and the circadian rhythm determines sleep–wake states in mammals. Disruption of the circadian rhythm usually causes sleep disorders [134]. VIP, a neurotransmitter expressed by a subset of the SCN neurons, appears to play a critical role in the regulation of the circadian rhythm and sleep–wake behaviors [135,136,137]. VIPergic neurons project densely throughout the SCN. A previous study revealed that VIP kickout mice showed an 8-h advance of the predicted activity phase with less precision when exposed to constant darkness [138]. This shows that VIP and its receptors are critical for maintaining normal circadian rhythms, which is a significant function of the SCN.

Previous pharmacological studies have shown that VIP is involved in sleep regulation, especially in stimulating REM sleep. For example, the intracerebroventricular administration of VIP in rats, rabbits, and cats resulted in a significant increase in the duration of REM sleep [139,140]. Injection of a competitive VIP receptor antagonist in rats reduced the total time of REM sleep as well [141]. Many recent studies have indicated that mice lacking VIP or VIP receptor-2 show a markedly disrupted circadian rhythm [142,143]. Moreover, when VIP was microinjected into the pontine reticular tegmentum and the oral pontine tegmentum, REM sleep was enhanced chronically in rats [144]. Collins et al. found that silencing the VIPergic neurons in the SCN suppressed nighttime but not daytime sleep [145]. Overall, these findings suggest that VIP can be applied to improve sleep, especially REM sleep.

Previous studies on the effect of VIP on sleep–wake regulation and circadian rhythms are not clear, and the neurobiological mechanisms are also unknown. Further research needs to be conducted.

### 2.8. Other Neuropeptides Associated with Sleep and Wake Regulation

Previous studies have shown that neuropeptides have diverse effects on sleep–wake regulation in rodents and humans. Here, we summarize the roles of other neuropeptides in sleep–wake regulation. Adrenocorticotropic hormone, cocaine- and amphetamine-regulated transcript, ghrelin, neurotensin, pituitary adenylyl cyclase-activating polypeptide, and somatostatin induce arousal [146,147,148,149,150,151]. Some of them (neurotensin, pituitary adenylyl cyclase-activating polypeptide, and somatostatin) tend to suppress NREM sleep, but they have a positive impact on REM sleep [148,149,150]. Other neuropeptides, including brain-derived neurotrophic factor, growth hormone, growth hormone-releasing hormone, interleukin 1 beta, leptin, melanocyte-stimulating hormone, neuropeptide B, opioid peptides, and tumor necrosis factor, can promote NREM sleep [152,153,154,155,156,157,158]. Cholecystokinin has been shown to play a complex role in the regulation of sleep–wake behaviors, because it induces both NREM sleep and wakefulness [154]. Recent studies have also suggested that both cholecystokinin and transforming growth factor alpha can potentially inhibit the transition from sleep to wakefulness [159].

## 3. Conclusions

Neuropeptides are widely distributed in the brain and are involved in the regulation of many physiological functions, including sleep–wake behaviors. Physiological behaviors and circadian regulation of sleep vary from species to species, and the different expression levels of neuropeptides in various animals may explain the difference. In this review, we summarized the functions of different neuropeptides in the regulation of sleep–wake behaviors (Table 1). Wakefulness-promoting neuropeptides mainly include orexin and NPS, while galanin, MCH, and VIP have a positive impact on sleep, especially REM sleep. NPY and SP tend to have a dual effect on sleep–wake behaviors.

Although previous studies have found some of the mechanistic links between sleep–wake behaviors and autonomic regulation, the neurobiological mechanisms are still poorly understood. Various neuropeptides are highly involved in homeostatic regulation. Their presence in sleep- and/or wakefulness-promoting neurons may facilitate the coupling of sleep–wake behaviors with autonomic regulation. Thus, further studies on neuropeptides will not only uncover the complex control network of sleep–wake regulation but also explain the association between sleep and autonomic responses.

In addition, further studies on neuropeptides are beneficial to explain the pathogenesis of sleep disorders. For example, a recent study has found that traumatic brain injury leads to a reduction in orexin-A-positive neurons in rodents. The injured rodents are unable to maintain wakefulness and suffer from cognitive impairment as well as memory loss. Given that orexin maintains wakefulness, the decreased orexin neurons may explain why traumatic brain injury patients tend to suffer from sleep disorders, including hypersomnia and excessive daytime sleepiness.

Neuropeptides and their receptor antagonists also have clinical implications, as these chemicals exert specific effects on sleep–wake behaviors. As mentioned above, neuropeptide agonists might serve as novel therapeutics to treat sleep disorders and anxiety neurosis. Furthermore, orexin administration stabilizes wakefulness and reduces sleep–wake fragmentation, which is commonly observed in Alzheimer’s patients. Therefore, the neuropeptide family has wide applications in the treatment of sleep disorders and other mental diseases.

In summary, neuropeptides in the CNS play an important regulatory role in sleep–wake behaviors. An in-depth understanding of neuropeptides will help us further explore the mysteries of the CNS.

## Figures and Tables

**Figure 1 ijms-23-04599-f001:**
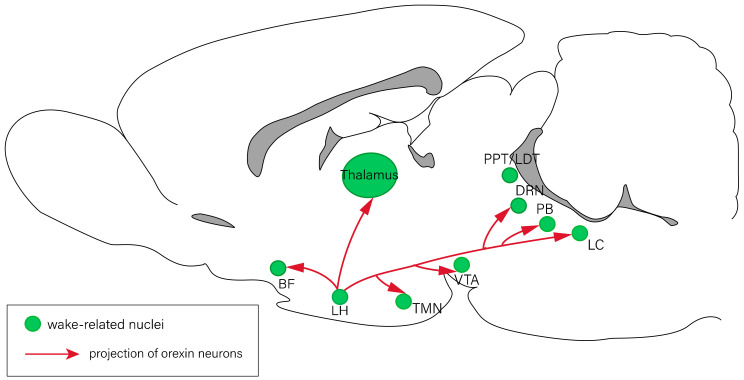
Neural circuits of arousal regulation. The nuclei related to arousal response include the BF, LH, TMN, VTA, PPT/LDT, DRN, PB, and LC. Red lines represent the projections of orexin neurons in the LH. BF: basal forebrain; DRN: dorsal raphe nucleus; LC: locus coeruleus; LDT: laterodorsal tegmental nucleus; LH: lateral hypothalamus; PB: parabrachial nucleus; PPT: pedunculopontine tegmental nucleus; TMN: tuberomammillary nucleus; VTA: ventral tegmental area.

**Figure 2 ijms-23-04599-f002:**
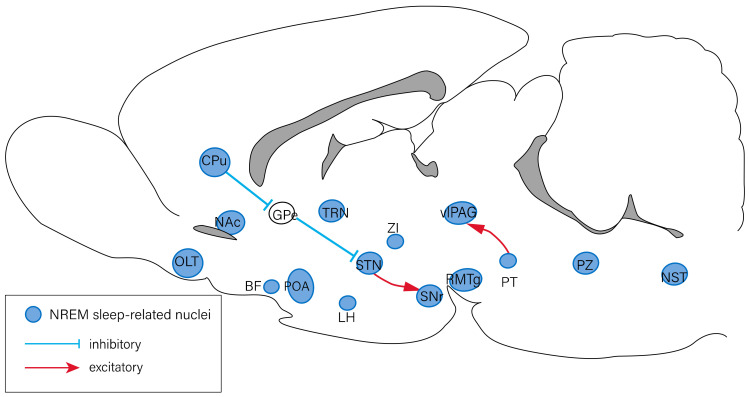
Neural circuits of NREM sleep regulation. The nuclei involved in the regulation of NREM sleep include the OLT, CPu, NAc, GPe, BF, POA, TRN, LH, STN, ZI, SNr, RMTg, vlPAG, PT, PZ, and NST. The PT has an excitatory projection to the vlPAG, and the CPu inhibits the STN neurons by suppressing the GPe. The STN has an excitatory projection to the SNr. BF: basal forebrain; CPu: caudate putamen; GPe: external globus pallidus; LH: lateral hypothalamus; NAc: nucleus accumbens; NST: nucleus of solitary tract; OLT: olfactory tubercles; POA: preoptic area; PT: pontine tegmentum; PZ: parafacial zone; RMTg: rostromedial tegmental nucleus; SNr: subtantia nigra pars reticulata; STN: subthalamic nucleus; TRN: thalamic reticular nucleus; vlPAG: ventrolateral periaqueductal gray; ZI: zona incerta.

**Figure 3 ijms-23-04599-f003:**
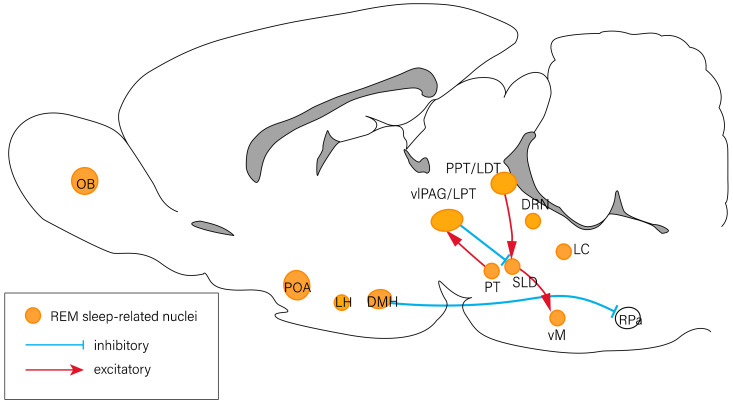
Neural circuits of REM sleep regulation. The OB, POA, LH, DMH, PT, vlPAG/LPT, PPT/LDT, DRN, LC, and vM are associated with the modulation of REM sleep. There exist inhibitory projections (blue lines) and excitatory projections (red lines) between nuclei. DMH: dorsomedial hypothalamus; DRN: dorsal raphe nucleus; LC: locus coeruleus; LDT: laterodorsal tegmental nucleus; LH: lateral hypothalamus; LPT: lateral pontine tegmentum; OB: olfactory bulb; POA: preoptic area; PPT: pedunculopontine tegmental nucleus; PT: pontine tegmentum; RPa: raphe pallidus area; vlPAG: ventrolateral periaqueductal gray; vM: ventral medulla.

**Table 1 ijms-23-04599-t001:** The influence of neuropeptides on sleep–wake states and underlying neurobiological mechanisms.

Neuropeptide	Distribution	Type	Subject	Effects on Sleep–Wake Cycle	Nuclei Involved in the Regulation of the Sleep–Wake Cycle
Galanin	Throughout the CNS	Intravenous galanin;	Humans	REM sleep (+) [37]	GABAergic neurons in the VLPO [39]; noradrenergic neurons in the LC [40]
		Overexpression of galanin	Zebrafish	Sleep (+), locomotor activity (−) [38]	
Orexin	Peripherical area of the lateral hypothalamus	Intracerebroventricular administration of orexin-A;	Mice	NREM sleep (−), REM sleep (−), Wake (+) [51]	Monoaminergic neurons in the LC [36,37], DRN [37,38], TMN [25,39]; cholinergic neurons in the BF and LDT [40,41]; GABAergic neurons in the POA [43]; 5-HT neurons in the DRN [44,48]; cholinergic neurons in the BF [45]
		Application of orexin-A into LC, TMN, DRN, BF, and LDT;	Mice	Wake (+)	
		Orexin receptor-2 antagonist	Mice/humans	NREM sleep (+), REM sleep (+) [52]	
Melanin-concentrating hormone	Lateral hypothalamus and zona incerta	Administration of MCH;	Rats	NREM sleep (+), REM sleep (+) [59]	VLPO; vlPAG and the lateral pontine tegmentum [65]
		Intracerebroventricular administration of MCH receptor-1 antagonist	Rats	NREM sleep (−), REM sleep (−) [60]	
Neuropeptide S	Brainstem and the limbic system	Intracerebroventricular administration of NPS;	Rats	Wake (+) [76]	Sleep-promoting neurons in the VLPO [77]
		Infusion of NPS	Rats	NREM sleep (−), Wake (+) [77]	
Neuropeptide Y	Hypothalamus	Application of NPY;	Humans	Sleep (+) [88]	Paraventricular nucleus [81]; VLPO [94]
		Intraventricular injection of NPY;	Rats	Sleep (+) [90]	
		Application of an NPY receptor-2 agonist;	Rats	NREM sleep (+) [92]	
		Injection of NPY into the paraventricular nucleus	Rats	Sleep (−) [81]	
Substance P	Throughout the CNS	Microinjection of SP into VLPO;	Rats	NREM sleep (+) [102]	VLPO; cerebral cortex; histaminergic neurons in the TMN [107]
		Intracerebroventricular administration of SP;	Mice	NREM sleep (+) [104]	
		Administration of SP	Mice/humans	Wake (+) [105]	
Vasoactive intestinal peptide	Gut, pancreas, and SCN	Intracerebroventricular administration of VIP;	Rats/rabbits/cats	REM sleep (+) [115,116]	VIPergic neurons in the SCN [121]
		Injection of a VIP receptor antagonist	Rats	REM sleep (−) [117]	

(+), increased; (−), decreased.

## Data Availability

Not applicable.

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
