# Peer review of "Roles of Neuropeptides in Sleep–Wake Regulation"

_ijms, 2022, doi:10.3390/ijms23094599_

Round 1
Reviewer 1 Report
The review on the “Roles of Neuropeptides in Sleep and Wake Regulation: Review of Recent Findings” very briefly present documents some findings on the involvement of some sleep-promoting peptides and inducing wakefulness. In the current organization the review appears like a list of selected studies focused on the involvement of different neuropeptides in promoting sleep and/or arousal. Many critical information are lacking from the main structure, for instance limitations of experimental designs through which certain physiological and pharmacological actions are demonstrated and/or hypothesized. Although the issue the manuscript addresses ir relevant, I do not consider the review acceptable in the current form. I would suggest to conduct a more critical review. For instance, to identify a key specific moment (ie. sleep induction, NREM) and review the actions of the different neuropeptides, to propose a strong and robust the state of the art for advancing in this research field.
Reviewer 2 Report
This is a balanced review of the evolving complexity of the neuropeptides in sleep-wake regulation. However, there is still minor concern regarding the accuracy of some statements in this review. Please review my suggestions.
- Abstract: The outlined scope and aims were reasonably and well defined.
- The authors stated the importance of the sleep-wake cycle in mammals. Please consider a short sentence about the critical effects of sleep loss in the Background.
- A more detailed definition of the regulation of neuropeptides (synthesis, storage, and secretion) is missing in the Introduction section.
- Also, in order to provide a balanced review, the authors should add a comment regarding the function of proteases in the regulation of activation or/and inactivation of neuropeptides.
- Some information is missing within these sentences:
Line 121 “A study on mice lacking 5-HT receptors has shown an increase in REM sleep [19]”. Make clear that in this study, mice lacking 5-HT receptors, specifically in orexin neurons, have shown an increase in REM sleep.
Line 134 “However, orexin also activates the local GABAergic input to orexin neurons. Genetic disruption of the GABAergic input has been reported to produce abnormal sleep-wake states [52]”. Ablation of GABAergic input in orexin-producing neurons has been reported to produce changes in sleep-wake states.
- Line 122 “Ablation of 5-HT neurons in the CNS decreased the duration of REM sleep and attenuated the arousal response [49]”. I don’t see the connection of this study (ref 49) with the orexin receptor. Please comment.
- Authors may add a sentence about the possible role of opioid peptides in sleep regulation and potential clinical importance to their conclusions.
- As evidenced by the references, only a few recent papers have been published in this area, which indicates a lack of significant advances in this field. Hence, the title of this paper should be re-evaluated.
